# Effects of Dietary Inclusion of Enzymatically Hydrolyzed Compound Soy Protein on the Growth Performance and Intestinal Health of Juvenile American Eels (*Anguilla rostrata*)

**DOI:** 10.3390/ani14213096

**Published:** 2024-10-27

**Authors:** Yichuang Xu, Chengyao Wu, Pan Wang, Xiaozhao Han, Jinyue Yang, Shaowei Zhai

**Affiliations:** 1Fisheries College, Jimei University, Xiamen 361021, China; xuyichuang@jmu.edu.cn (Y.X.); 54chengyaowu@sina.com (C.W.); 202311908028@jmu.edu.cn (P.W.); 202414908011@jmu.edu.cn (X.H.); 2Engineering Research Center of the Modern Technology for Eel Industry, Ministry of Education of China, Xiamen 361021, China; 3College of Ocean Food and Biological Engineering, Jimei University, Xiamen 361021, China; 202461000113@jmu.edu.cn

**Keywords:** enzymatically hydrolyzed soy protein, growth performance, intestinal health, American eels

## Abstract

The drive to diminish reliance on fishmeal and reduce feed costs has pushed the development of fishmeal alternatives. In recent years, the use of enzymatically hydrolyzed soybean as a fishmeal substitute in aquafeeds has garnered growing attention and application. This study demonstrated that appropriate levels of dietary enzymatically hydrolyzed compound soy protein (EHCS), as a substitute for white fishmeal, significantly enhanced growth performance and improved the intestinal health of American eels. Based on the quadratic regression analysis of weight gain rate and feed efficiency against the dietary EHCS inclusion levels, the optimal inclusion levels for American eels ranged from 7.78% to 8.33% with maximum levels spanning 17.59% to 17.77%, which corresponded to white fishmeal replacement levels of 24.31–26.03% and 54.97–55.53%, respectively. To ensure normal growth performance, it is recommended that the inclusion level of EHCS should not exceed 17.77%. Our findings offer a valuable reference for the application of EHCS as a white fishmeal substitution in the diets of American eels.

## 1. Introduction

Fishmeal is an optimal protein source in aquafeeds due to its high protein content, well-balanced amino acid composition, and excellent palatability [1]. Nevertheless, the escalating scarcity of marine fishery resources coupled with the rapid development of aquaculture have led to a supply shortage of fishmeal and a consequent rise in fishmeal prices, which raised the cost of aquafeeds [2,3]. Therefore, the quest to reduce reliance on fishmeal and lower feed costs have propelled the development of cost-effective fishmeal substitutions to the forefront of research in the aquaculture [4,5,6].

Among the numerous protein sources, soybean is commonly utilized as a fishmeal substitution in aquafeeds owing to its ideal protein content, rich amino acid profile, abundant availability, and low cost [7,8]. However, untreated soybean contains various antinutritional factors that not only impair protein digestibility for aquatic animals but also have adverse effects on the health of aquatic animals, limiting their use in aquafeeds [7,8]. Therefore, the elimination of antinutritional factors in soybeans is advantageous for their application in aquafeeds. Enzymatically hydrolysis is particularly noteworthy as a technique employed to reduce antinutritional factors due to its mild conditions and high efficiency, which result in the effective removal of antinutritional factors and minimal nutrient loss in soybean [6,9]. Additionally, the large molecular proteins, starch, cellulose, and mannans present in soybeans undergo enzymatic hydrolysis into smaller peptides, amino acids, glucose, maltose, and other nutrients that are readily absorbable [10]. Therefore, compared with untreated soybean, enzymatically hydrolyzed soybean exhibits a significant reduction in antinutritional factor content and an enhanced nutritional value.

In recent years, the utilization of enzymatically hydrolyzed soybean as a fishmeal substitute in aquafeeds has seen increasing attention and application [4,5,6]. Huang et al. [6] demonstrated that replacing up to 45% of dietary fishmeal with enzymatically hydrolyzed soybean did not adversely impact the growth performance of juvenile Chinese mitten crabs (*Eriocheir sinensis*). Tibaldi et al. [4] also revealed that substituting 50% of dietary fishmeal with enzymatically hydrolyzed soybean meal has no adverse effects on the growth performance and whole-body composition of European sea bass (*Dicentrarchus labrax*). Moreover, hydrolyzing soybean using a multi-enzyme strategy exhibits a higher efficacy in substituting fishmeal in aquafeeds compared to its single-enzyme counterpart [5]. It was reported that protease-treated soybean meal could substitute for 20% of fishmeal in the diets of largemouth bass (*Micropterus salmoides*), whereas a combination treatment with protease and non-starch polysaccharide enzymes could facilitate soybean meal replacing up to 47.27% of dietary fishmeal for largemouth bass [5]. Therefore, soybean protein hydrolyzed through multiple enzymes represents a highly efficacious substitute for fishmeal in aquafeeds.

Eels (*Anguilla* spp.) are widely distributed throughout the world, and American eels (*Anguilla rostrata*) represent one of the primary cultured eel species in southern China. Eels have a high protein requirement, and the highest-grade white fishmeal traditionally serves as the primary protein source in their diets. Due to the escalating costs of fishmeal, there has been a significant focus on investigating substitutes for fishmeal in eel diets. Considering that the potential of soybean hydrolyzed by a combination of multiple enzymes to replace fishmeal in feed, this study was carried out to assess the effects of substituting dietary white fishmeal with enzymatically hydrolyzed compound soy protein (EHCS), treated with a combination of multiple enzymes, on the growth performance and intestinal health of juvenile American eels. Our findings are anticipated to offer valuable insights into the application of EHCS in the diets for American eels, as well as potentially for other fish species, to reduce the feed cost and promote the development of low/non-fishmeal diets in the near future.

## 2. Materials and Methods

### 2.1. Ethics Statement

This study on American eels culture and management adhered to the Management Rule of Laboratory Animals (Chinese Order No. 676 of the State Council, revised 1 March 2017). All necessary measures were taken to minimize animal suffering. The Jimei University Ethics Committee approved the research protocols (identification code: 2021-11 and approved in 11 March 2021).

### 2.2. Diet Preparation and Fish Culture

Five isonitrogenous and isolipidic experimental diets were prepared, incorporating enzymatically hydrolyzed compound soy protein (EHCS) at graded levels of 0% (EHCS0), 8% (EHCS8), 16% (EHCS16), 24% (EHCS24), and 32% (EHCS32), corresponding to progressive replacement levels of white fishmeal at 0%, 25%, 50%, 75%, and 100%, respectively (Table 1). The EHCS was sourced from Xiamen Longli Industry and Trade Co., Ltd., Xiamen, China. The EHCS comprised a mixture of soybean meal, soy protein concentrate, and soy protein isolate in a 4:5:1 ratio, which was enzymatically hydrolyzed with alkaline protease (750 IU/g substrate), neutral protease (250 IU/g substrate), and acid protease (250 IU/g substrate) at a controlled temperature of 4 °C for a duration of 24 h. For the preparation of experimental diets, the ingredients were first crushed using a feed grinder (ZFJ-300, Jiangyin Ruizong Machinery Manufacturing Co., Ltd., Wuxi, China); then, they were sieved through an 80 μm mesh, thoroughly mixed, and stored at −20 °C until feeding. Prior to feeding, the powdered diet was hydrated with water at a ratio of 1:1.2 to attain a dough-like consistency. The amino acid composition of the experimental diets and the levels of antinutritional factors in EHCS are detailed in Table 2 and Table 3, respectively.

After a 4-week acclimatization period, a total of 500 health American eels (mean initial weight: 26.00 ± 0.02 g per fish, mean ± S.D.) were randomly assigned to the EHCS0 group, EHCS8 group, EHCS16 group, EHCS24 group, and EHCS32 group, respectively. Each tank contained 25 fish with four replicates per group. The feeding trial lasted for 10 weeks. An automatic temperature control system with flow through was utilized throughout the trial. All fish were fed to satiation twice daily at 07:00 and 17:00. The uneaten feed was collected to assess the feed consumption. The water quality parameters were monitored manually twice a week and are detailed as follows: temperature 25.3–27.5 °C, pH 7.2–8.2, dissolved oxygen 6.5–8.5 mg/L, total ammonia nitrogen ≤ 0.12 mg/L, nitrite ≤ 0.02 mg/L.

### 2.3. Sample Collection

At the end of the feeding trial, the eels were fasted for 24 h. The number and weight of eels in each tank was recorded. Eels were anesthetized using eugenol 50 mg/L in 1:4 dilution with rearing water for about 5 min. The collected serum samples were stored at −80 °C for subsequent biochemical analysis. Anterior intestine samples from four eels per tank were sampled and rapidly frozen in liquid nitrogen and stored at −80 °C for determining intestinal digestive enzyme activity and antioxidant-related indices. Anterior intestine samples from two eels per tank were fixed in Bouin’s solution and stored at 4 °C for morphological observation. The anterior intestines of six eels per tank were rapidly frozen in liquid nitrogen and then stored at −80 °C for intestinal microbiota analysis. Ten eels from each tank were randomly selected for whole-body composition analysis.

### 2.4. Growth Performance

The final body weight (FBW), weight gain rate (WGR), specific growth rate (SGR), feed intake (FI), feed efficiency (FE), feed conversion ratio (FCR), protein efficiency ratio (PER), and survival rate (SR) were calculated in accordance with methodologies described in previous studies [6,11].

### 2.5. Proximate Composition, Amino Acid Composition, and Antinutritional Factors

The moisture content of whole-body fish and experimental diets samples were measured by a freeze-dryer (Labogene, Lillerød, Denmark). The contents of crude protein, crude lipid, ash, calcium (Ca), and phosphorus (P) were evaluated based on the description of the AOAC [12]. The levels of trypsin inhibitor factor (TIF, #KTI-EA), glycinin (#11S-EA), and β-conglycinin (#7S-EA) were analyzed with the utilization of kits (Tianjin Longke Xinyu Biotechnology Co., Ltd., Tianjin, China). The amino acid composition was measured according to the “Determination of amino acids in feeds” (GB/T 18246-2019) published by the Standardization Administration of China in 10 December 2019 [13].

### 2.6. Serum Biochemical Indexes

Alkaline phosphatase (AKP, #A059-2-2), complement 3 (C3, #H186-1-2), diamine oxidase (DAO, #A088-2-1), D-lactic acid (D-lac, #A019-3-1), glutamic-oxalacetic transaminase (GOT, #C010-1-1), glutamic-pyruvic transaminase (GPT, #C009-1-1), high-density lipoprotein cholesterol (HDL-C, #A112-1-1), immunoglobulin M (IgM, #A109-1-2), low-density lipoprotein cholesterol (LDL-C, #A113-1-1), lysozyme (LZM, #A050-1-1), total cholesterol (TC, #A111-1-1), and triglyceride (TG, #A110-1-1) were determined with the utilization of kits (Nanjing Jiancheng Bioengineering Co., Ltd., Nanjing, China).

### 2.7. Activities of Intestinal Digestive Enzymes

The intestinal protease activity was evaluated using the Folin phenol reagent. The intestinal activities of amylase (#C016-1-1) and lipase (#A0542-1) were measured using commercial kits (Nanjing Jiancheng Bioengineering Co., Ltd.).

### 2.8. Intestinal Antioxidant Parameters

The intestinal activities of superoxide dismutase (SOD, #A001-1-1), catalase (CAT, #A007-1-1), and glutathione peroxidase (GSH-PX, #A005-1-2), as well as the total antioxidant capacity (T-AOC, #A015-2-1) and the level of malondialdehyde (MDA, #A003-1-2), were quantified using commercial assay kits (Nanjing Jiancheng Biotechnology Co., Ltd., Nanjing, China).

### 2.9. Histological Observation

The preparation of intestinal tissue sections was carried out based on the methodology described by Xu et al. [14]. The tissue sections were stained with hematoxylin and eosin (H&E) and subsequently observed under a microscope (BX80-JPA, Olympus, Tokyo, Japan) for capturing representative micrographs. Morphometric analysis was conducted using Image-Pro Plus 6.0 software (Media Cybernetics, Silver Spring, MD, USA).

### 2.10. Intestinal Microbiota Profiling

The intestinal microbiota analysis was based on the description from Lu et al. [15]. In summary, total bacterial DNA was extracted from American eel intestinal samples. The forward (338F, 5′-ACTCCTACGGGAGGCAGCAG-3′) and reverse (806R, 5′-GACTACHVGGGTWTCTAAT-3′) primers were employed for PCR amplification of the V3-V4 hypervariable region of the bacterial 16S rRNA gene. High-throughput sequencing was then conducted using the Illumina Miseq PE300 platform (Beijing Allwegene Tech. Co., Ltd., Beijing, China).

### 2.11. Statistical Analysis

The data were presented as mean ± standard deviation (S.D., *n* = 4). One-way ANOVA was performed using SPSS 26.0 (SPSS, Chicago, IL, USA), and Duncan’s method was used for multiple comparisons. Differences were considered significant at *p* < 0.05. When the analyzed data were expressed as percentages, and the overall rate was less than 30% or greater than 70%, a square root arcsine transformation was applied before statistical analysis. QIIME (v1.8.0) software was used to analyze the alpha diversity indices of the intestinal microbiota. PLS-DA was used for beta diversity analysis, and Lefse analysis was conducted using Python v2.7 software with an LDA score threshold of 3 and *p* < 0.05.

## 3. Results

### 3.1. Growth Performance

As shown in Table 4, the WGR, SGR, FE, and PER values were higher for American eels fed the EHCS8 diet than those fed the EHCS0 diet. There were no significant differences in the WGR, SGR, FE, FCR, and PER values between American eels fed the EHCS16 diet and the EHCS0 diet (Table 4). American eels fed the EHCS24 and EHCS32 diets showed lower WGR, SGR, FE, FCR, and PER values compared to those fed the EHCS0 diet (Table 4). American eels fed the EHCS32 diet also exhibited lower SRs than those fed other four diets (Table 4). Based on the quadratic regression analysis of WGR and FE against the dietary EHCS inclusion levels, the optimal levels of dietary EHCS inclusion were 7.78% and 8.33%, respectively, while the maximum levels that exhibited comparable effect of the diet without EHCS inclusion were 17.77% and 17.59%, respectively, for American eels (Figure 1A,B).

### 3.2. Body Composition

As shown in Table 5, the different dietary inclusion levels of EHCS did not affect the levels of moisture, crude protein, crude lipid, ash, calcium, and phosphorus in the whole body of juvenile American eels.

### 3.3. Serum Biochemical Parameters

The effects of dietary EHCS inclusion on the serum biochemical parameters of American eels are presented in Figure 2. The serum activities of GOT, GPT, and DAO, along with the serum levels of TG and D-lac, initially decreased and then increased as the dietary inclusion levels of EHCS increased (Figure 2). American eels fed the EHCS8 diets exhibited the lowest serum activities of GOT, GPT, and DAO, and the lowest serum contents of TG and D-lac, compared to those fed the other four diets (Figure 2). The serum activities of AKP and LZM, alongside the serum levels of lgM and C3, initially increased and subsequently decreased as dietary inclusion levels of EHCS increased (Figure 2). American eels fed the EHCS8 diet exhibited the highest serum activities of AKP and LZM, along with the serum levels of lgM and C3, compared to those fed the other four diets (Figure 2). The serum TC content was higher, and the serum HDL-C content was lower, for eels fed the EHCS24 and EHCS32 diets with no significant differences among eels fed the other three diets (Figure 2). The serum LDL-C content was the highest for eels fed the EHCS32 diets and showed no significant differences among the eels that were fed the other four diets (Figure 2).

### 3.4. Intestinal Histology

The effects of dietary EHCS inclusion on the intestinal histology are illustrated in Figure 3. American eels fed the EHCS8 diet exhibited increased muscular thickness (MT), whereas those fed the EHCS24 and EHCS32 diets exhibited decreased MT and villus height (VH) compared to those fed the EHCS0 diet (Figure 3). There was no significant difference in MT and VH between American eels fed the EHCS0 and EHCS16 diets (Figure 3).

### 3.5. Intestinal Digestive Enzymes Activities

The effects of dietary EHCS inclusion on the intestinal digestive enzyme activities are illustrated in Figure 4. American eels fed the EHCS8 and EHCS16 diets showed no significant differences in lipase and protease activities, whereas those fed the EHCS24 and EHCS32 diets exhibited lower lipase and protease activities compared to those fed the EHCS0 diet (Figure 4A,B). Different dietary inclusion levels of EHCS had no effects on amylase activity (Figure 4C).

### 3.6. Intestinal Antioxidant Capacity

The impacts of dietary EHCS inclusion on the intestinal antioxidant capacity are shown in Figure 5. American eels fed the EHCS8 diet exhibited elevated T-AOC levels and increased activities of CAT, T-SOD and GSH-PX, while those fed the EHCS24 and EHCS32 diets exhibited reduced T-AOC levels and diminished activities of CAT, T-SOD and GSH-PX compared to those fed the EHCS0 diet (Figure 5A–D). American eels fed the EHCS8 and EHCS16 diets showed no significant differences in MDA content, whereas those fed the EHCS24 and EHCS32 diets exhibited elevated MDA content in comparison to those fed the EHCS0 diet (Figure 5E).

### 3.7. Intestinal Microbiota

Considering that American eels fed the EHCS8 diet exhibited superior growth performance and intestinal health, while those fed the EHCS32 diet experienced growth arrest and intestinal injury compared to those fed the EHCS0 diet, an analysis of intestinal microbiota was carried out among American eels fed these three diets to examine whether different dietary EHCS inclusion levels affect the microbiota composition of American eels. As shown in Figure 6A, there was no significant difference in the alpha diversity of the intestinal flora among eels fed the EHCS0, EHCS8, and EHCS32 diets. PLS-DA analysis revealed that the beta diversity of the intestinal microbiota among eels fed the EHCS0, EHCS8, and EHCS32 diets was dispersed, indicating differences in intestinal microbiota composition among the three groups (Figure 6B). Compared to American eels fed the EHCS0 diet, the relative abundance of Cyanobacteria was increased in those fed the EHCS8 diet, and no difference was observed between eels fed the EHCS0 and EHCS32 diets (Figure 6C,D). Compared to American eels fed the EHCS0 diet, the relative abundances of *Paraclostridium*, *Serratia*, and *Ralstonia* were significantly decreased in those fed the EHCS8 and EHCS32 diets; the relative abundances of *Chryseobacterium*, *Acidithiobacillaceae__KCM_B_112*, and *Pseudomonas* were significantly increased in those fed the EHCS8 diet; and the relative abundance of *Brevundimonas* was significantly increased in those fed the EHCS32 diet (Figure 6E). These findings suggest that different dietary inclusion levels of EHCS significantly modulate the gut microbial community structure in American eels.

## 4. Discussion

### 4.1. Growth Performance

Previous studies on substituting fish meal with enzymatically hydrolyzed soybean protein sources in aquafeeds primarily focused on single protein sources with limited research on the replacement of fish meal with EHCS in the diet of fish. The present study demonstrated that the WGR, SGR, FE, and PER of American eels initially increased and subsequently decreased with incremental levels of EHCS inclusion. Moreover, this study determined that the optimal and maximum levels of dietary EHCS inclusion ranged from 7.78% to 8.33% and from 17.59% to 17.77%, respectively. It was also revealed that appropriate levels of dietary enzymatically hydrolyzed soybean protein as a replacement for fishmeal promoted growth performance, whereas excessive levels inhibited growth in other fish species, including Japanese flounder (*Paralichthys olivaceus*) and starry flounder (*Platichthys stellatus*) [16,17]. An enzymatic hydrolysis of soy protein has been shown to increase the proportions of small peptides and free amino acids, enhancing the nutritional utilization and growth performance of aquatic animals [6,18,19]. However, excessive levels of enzymatically hydrolyzed soybean protein inclusion would lead to an imbalanced amino acid composition and poor palatability of the diet, which adversely affect the growth of fish [16]. Although enzyme treatment reduces the contents of antinutritional factors in enzymatically hydrolyzed soybean protein, increasing its dietary levels still results in high contents of antinutritional factors, ultimately suppressing the growth of fish [19]. Taken together, EHCS is suitable as fishmeal substitute in diets for the growth of American eels and other aquatic animals when used at moderate levels.

### 4.2. Body Composition

This study demonstrated that increasing levels of dietary EHCS inclusion did not result in significant changes in the whole-body composition of American eels. Research on the effects of EHCS inclusion in fish feed on body composition is limited at present. Previous studies indicated that substituting dietary fish meal with enzymatically hydrolyzed soybean meal did not lead to significant alterations in the whole-body composition of flounder [17]. However, high levels of dietary enzymatically hydrolyzed soy protein replacing fishmeal were shown to significantly increase moisture levels and markedly decrease ash levels in the whole body of totoaba (*Totoaba macdonaldi*) [18]. Therefore, the impact of different dietary sources of enzymatically hydrolyzed soy protein inclusion on the proximate composition of the whole-body of fish exhibits significant variability.

### 4.3. Serum Biochemical Parameters

Increased serum GOT and GPT activities commonly serve as pivotal signs of liver damage [20]. Our study demonstrated that serum GOT and GPT activities decreased in response to the optimal level of dietary EHCS inclusion, indicating a hepatoprotective effect of optimal dietary EHCS levels. Conversely, excessive levels of dietary EHCS inclusion led to an increase in these two enzymes activities, suggesting a detrimental impact of excessive EHCS on liver function. Song et al. [21] reported that starry flounder fed diets with increasing levels of dietary enzymatically hydrolyzed soy protein replacing fishmeal exhibited analogous changes in serum GOT and GPT activities. Previous studies revealed that an appropriate level of dietary small peptides reduced the serum GOT and GPT activities [22]. Additionally, the enzymatic hydrolysis of soy protein has been shown to increase the proportions of small peptides [17], potentially contributing to the observed reduction in serum GOT and GPT activities of American eels. Our results showed that excessive levels of EHCS inclusion resulted in a deficiency of essential amino acids in the diet, which was shown to result in increased activities of serum GOT and GPT [23]. Furthermore, the antinutritional factors present in EHCS were not completely eliminated. Elevated levels of dietary EHCS inclusion led to an accumulation of these antinutritional factors, which could ultimately result in liver damage [24].

TC and TG are mainly synthesized in the liver and constitute essential components of serum lipids, while LDL-C and HDL-C are critical lipoproteins facilitating cholesterol transport within the body [25,26]. LDL-C is responsible for transporting cholesterol to peripheral tissues, while the primary function of HDL-C is to remove excess cholesterol and low-density lipoproteins from the bloodstream and cells [25]. In the current study, the serum levels of TG initially decreased and then increased as the level of dietary EHCS inclusion increased. The contents of TC and LDL-C were higher, and the HDL-C content was lower, for eels fed diets with excessive inclusion levels of EHCS. These findings suggested that American eels fed a diet with an optimal inclusion level of EHCS exhibited decreased levels of serum lipids and enhanced hepatic lipolysis, whereas an excessive inclusion level of EHCS disturbed hepatic lipid metabolism in American eels. Previous studies revealed that liver injury impaired its ability in lipid metabolism, which ultimately resulted in an increase in serum lipid levels [27,28]. The aforementioned results of serum GOT and GPT activities indicated that an optimal level of dietary EHCS enhanced liver function, while excessive levels of dietary EHCS impaired liver function. These changes in liver function likely contributed to the observed alterations in serum levels of TC, TG, LDL-C, and HDL-C in the present study. However, it was observed that starry flounder fed diets with progressively higher levels of dietary enzymatically hydrolyzed soy protein replacing fishmeal exhibited a reduction in the levels of TG, HDL-C, and LDL-C [17]. Thus, the tolerance of different fish species to dietary enzymatically hydrolyzed soy protein sources in terms of hepatic lipid metabolism varies significantly.

Serum AKP, IgM, C3, and LZM serve as crucial biomarkers for evaluating immune function. AKP contributes to the enhancement of nonspecific immunity [29]. IgM plays a pivotal role in the humoral immune response and is a critical immunoglobulin [30]. C3 not only possesses bactericidal and complement-activating properties but also functions in immune regulation [31]. LZM is involved in the clearance and phagocytosis of pathogenic microorganisms [30]. This investigation revealed that the LZM and AKP activities, along with the concentrations of lgM and C3, initially exhibited an uptrend, which was followed by a subsequent decline as the proportion of dietary EHCS inclusion increased. Therefore, American eels fed diets with an appropriate inclusion level of EHCS exhibited enhanced immunity, whereas an excessive application of EHCS detrimentally affected immune function. Additionally, it was evidenced that Chinese mitten crab (*Eriocheir sinensis*) fed diets wherein fishmeal was increasingly substituted with enzymatically hydrolyzed soybean exhibited similar alterations in immune function [6]. Prior investigations indicated that an optimal level of soybean peptides was able to enhance the nonspecific immunity and cellular viability in immune cells [32]. However, excessive levels of dietary EHCS inclusion resulted in the accumulation of antinutritional factors, which could ultimately lead to immune disorders [33]. Therefore, EHCS in moderate levels serves as a suitable substitute for white fishmeal in diets to support the immune function of American eels and other aquatic animals.

### 4.4. Intestinal Health

The intestinal tract is crucial for the absorption and digestion of nutrients [14,15]. The structural integrity and permeability, along with the activity of digestive enzymes and antioxidant capacity, serve as critical indicators for assessing intestinal health [14,15]. The thickening of the intestinal muscular layer and the presence of intestinal villi improve the efficiency of intestinal digestion and absorption of nutrients [14,15]. The intestinal lipase, protease, and amylase play a pivotal role in the digestion and absorption of proteins, lipids, and starch in the diets of fish [34]. Additionally, serum DAO activity and D-lac content are commonly utilized to evaluate intestinal mucosal permeability [14,35]. T-AOC serves as an important indicator for evaluating antioxidant capacity [36]. T-SOD, CAT, and GSH-P_X_ are key antioxidant enzymes responsible for scavenging oxygen free radicals in fish [14,37]. MDA is a lipid peroxidation by-product and a marker for oxidative stress [14,34]. This study demonstrated that American eels fed diets with an appropriate inclusion level of EHCS exhibited enhanced the structural integrity and antioxidant capacity of the intestines. It was revealed that soybean peptides exert protective effects on intestinal mucosa integrity and possessed chemical antioxidant activities, which may underlie the observed anxo-action of an optimal dietary EHCS inclusion level on the structural integrity and permeability, as well as antioxidant capacity, of intestinal tissue [38,39]. Our findings also indicated that excessive dietary EHCS inclusion levels are detrimental to the structural integrity, digestion and absorption of nutrients, and antioxidant capacity of American eels. Dan et al. [40] also reported that turbot juveniles fed diets with excessive replacement levels of fishmeal with α-galactosidase hydrolytic soybean meal showed decreases in MT and VH along with reduced intestinal digestive enzymes activities. Previous studies indicated that the antinutritional factors present in soybean and imbalanced amino acid composition were capable of reducing intestinal antioxidant capacity, leading to an overproduction of reactive oxygen species (ROS) and disruption of the intestinal mucosal structure [33,41,42,43,44]. Therefore, optimal levels of dietary EHCS inclusion enhanced the intestinal structure and antioxidant capacity, whereas excessive EHCS inclusion levels compromised it due to the accumulation of antinutritional factors present in EHCS and imbalanced amino acid composition in the diet.

The intestinal flora establishes a stable micro-ecosystem with the host, which performs critical physiological functions in nutrient metabolism, pathogen defense, and immune regulation, thus playing an indispensable role in growth and development [45,46]. In this study, compared to American eels fed the EHCS0 diet, those fed the EHCS8 diet exhibited a lower relative abundance of *Paraclostridium*, *Serratia*, and *Ralstonia*, alongside a higher relative abundance of *Chryseobacterium*, *Acidithiobacillaceae_KCM_B_112*, and *Pseudomonas*, in the intestines. *Paraclostridium* can reduce intestinal mucosal permeability, strengthen the mucosal barrier function, and prevent the colonization and proliferation of pathogenic bacteria [47]. *Serratia* is an opportunistic pathogen that poses a threat to the health of aquatic animals and is one of the primary pathogens responsible for diarrhea [48]. *Ralstonia* consists of various opportunistic pathogens, and its increased relative abundance can induce intestinal inflammation in the host [49]. *Chryseobacterium* and *Pseudomonas* were shown to promote nutrient digestion and absorption [50]. *Acidithiobacillaceae_KCM_B_112* is a member of the family *Acidithiobacillaceae* that is capable of activating elemental sulfur and converting it into bioavailable nutrients [51]. The above results indicate that the optimal dietary EHCS inclusion level is able to decrease the relative abundance of potential pathogenic bacteria while increasing the relative abundance of beneficial bacteria in the intestines of juvenile American eels, improving the intestinal microbiota balance, which may promote nutrient absorption and enhance immunity. Compared to American eels fed the EHCS0 diet, those fed the EHCS32 diet exhibited a lower relative abundance of *Paraclostridium*, *Serratia*, and *Ralstonia* but a higher relative abundance of *Brevundimonas* in the intestines. *Brevundimonas* is a potential pathogen capable of disrupting the intestinal microbiota balance and causing intestinal inflammation [52]. Although the above findings demonstrate that American eels fed diets with excessive EHCS inclusion exhibited an increased abundance of certain probiotics in the intestines compared with those fed the control diet, the aforementioned indicators related to intestinal health suggest that the probiotic effect of the elevated probiotics was insufficient to mitigate the intestinal damage resulting from the excessive dietary EHCS inclusion. Moreover, the elevated abundance of *Brevundimonas* may exacerbate the intestinal damage caused by excessive dietary EHCS inclusion.

In comparison with previous studies, the proportion of fishmeal substituted by enzymatically hydrolyzed soybean in this experiment was relatively modest. Firstly, the fishmeal substituted in this trial was premium-grade white fishmeal. Eels exhibit a stringent requirement for fishmeal quality in their diet, which presents challenges in substituting premium-grade fishmeal in eel diets. Furthermore, our findings indicate that an excessive inclusion of EHCS in eel diets could result in an imbalance in amino acid composition. This underscores the necessity of amino acid supplementation when utilizing EHCS to replace fishmeal in eel diets, ensuring that the amino acid composition optimally supports the growth and health of eels.

## 5. Conclusions

The current investigation unequivocally demonstrated that dietary EHCS inclusion significantly affects the growth performance and intestinal health of juvenile American eels. In comparison to eels fed diet without EHCS inclusion, those fed diet with EHCS inclusion levels at 8% exhibited superior growth performance and enhanced intestinal health. The diet containing a 16% inclusion level of EHCS had comparable effects on the growth performance and intestinal health of American eels compared to a diet without EHCS inclusion. However, dietary inclusion levels of EHCS reaching 24% were found to have negative impacts on both the growth performance and intestinal health of American eels. Furthermore, the 8% dietary inclusion level of EHCS was found to enhance the abundance of beneficial intestinal microbiota. Overall, it is recommended that the optimal levels of dietary EHCS inclusion should range from 7.78% to 8.33%, while the maximum permissible levels were suggested to be between 17. 59% and 17.77% for American eels. Our findings provide a valuable reference for the inclusion of EHCS in the diets of American eels as well as its potential applicability to other fish species, promoting the advancement of low or non-fishmeal diets.

## Figures and Tables

**Figure 1 animals-14-03096-f001:**
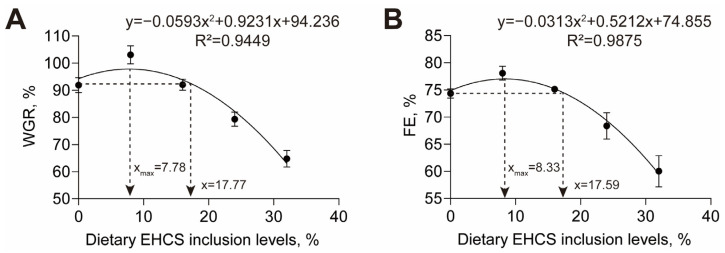
The relationship between dietary inclusion EHCS levels and the WGR (**A**) and FE (**B**) of juvenile American eels. Values are shown as mean ± S.D. (*n* = 4).

**Figure 2 animals-14-03096-f002:**
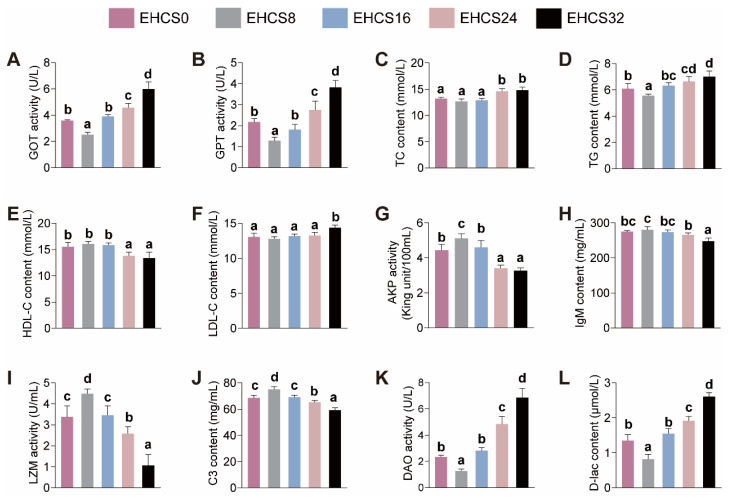
Effects of dietary EHCS inclusion on the serum biochemical parameters of juvenile American eels. (**A**) GOT activity. (**B**) GPT activity. (**C**) TC content. (**D**) TG content. (**E**) HDL-C content. (**F**) LDL-C content. (**G**) AKP activity. (**H**) lgM content. (**I**) LZM activity. (**J**) C3 content. (**K**) DAO activity. (**L**) D-lac content. Values are shown as mean ± S.D. (*n* = 4). Letters (a–d) denote significance at *p* < 0.05. AKP, acid phosphatase; C3, complement 3; D-lac, D-lactate; DAO, diamine oxide; GOT, glutamic-oxalacetic transaminase; GPT, glutamic-pyruvic transaminase; HDL-C, high-density lipoprotein cholesterol; IgM, immunoglobulin M; LDL-C, low-density lipoprotein cholesterol; LZM, lysozyme; TC, total cholesterol; TG, triglyceride.

**Figure 3 animals-14-03096-f003:**
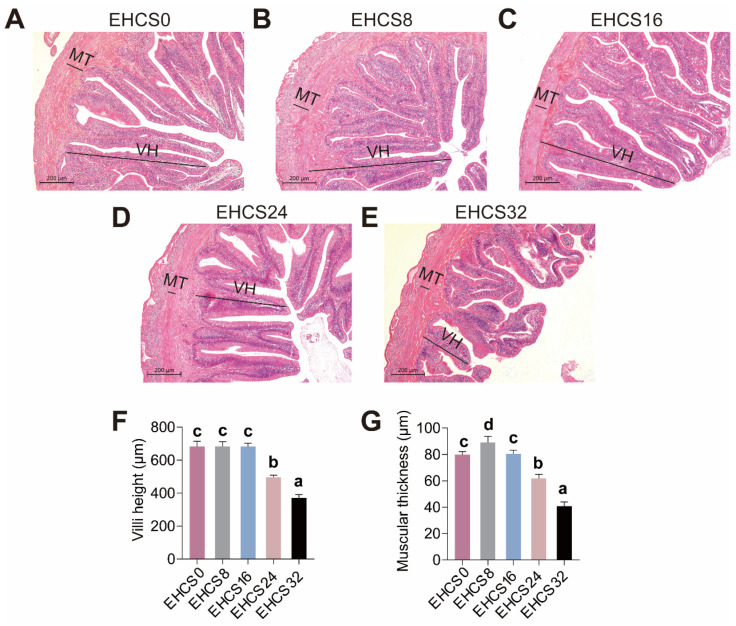
Effects of dietary EHCS inclusion on intestinal histology of juvenile American eels. (**A**–**E**) The representative photomicrographs of intestine sections of hematoxylin and eosin (H&E) staining, scale bar, 200 μm. VH, villus height. MT, muscular thickness. (**F**,**G**) The analysis of villus height and muscular thickness. Values are shown as mean ± S.D. (*n* = 4). Letters (a–d) denote significance at *p* < 0.05.

**Figure 4 animals-14-03096-f004:**
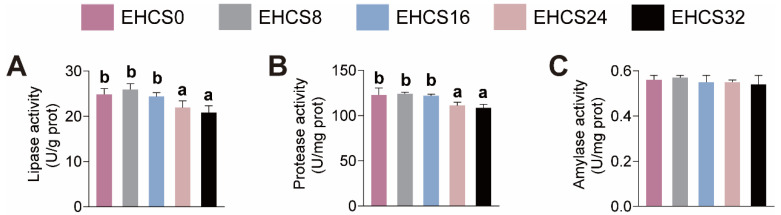
Effects of dietary EHCS inclusion on the activities of intestinal amylase (**A**), lipase (**B**), and proteases (**C**) activities of juvenile American eels. Values are shown as mean ± S.D. (*n* = 4). Letters (a–b) denote significance at *p* < 0.05.

**Figure 5 animals-14-03096-f005:**
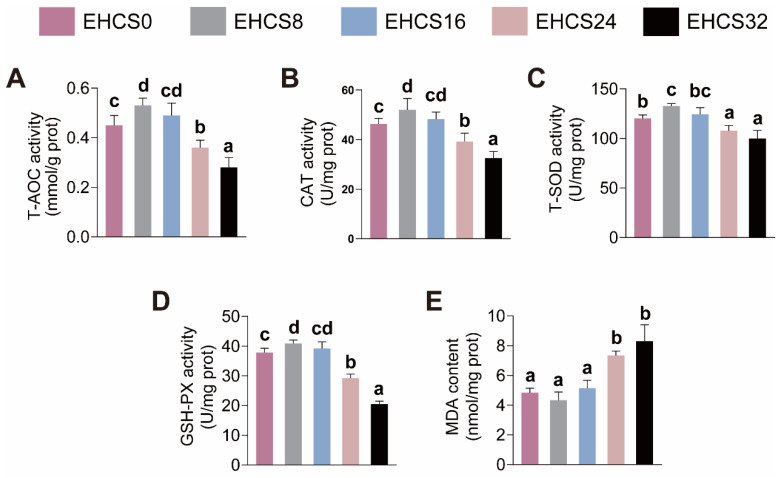
Effects of dietary EHCS inclusion on the intestinal antioxidant capacity of the juvenile American eels. (**A**) Total antioxidant capacity (T-AOC); (**B**) catalase (CAT) activity; (**C**) total superoxide dismutase (SOD) activity; (**D**) glutathione peroxidase (GSH-Px) activity; (**E**) malondialdehyde (MDA) level. Values are shown as mean ± S.D. (*n* = 4). Letters (a–d) denote significance at *p* < 0.05.

**Figure 6 animals-14-03096-f006:**
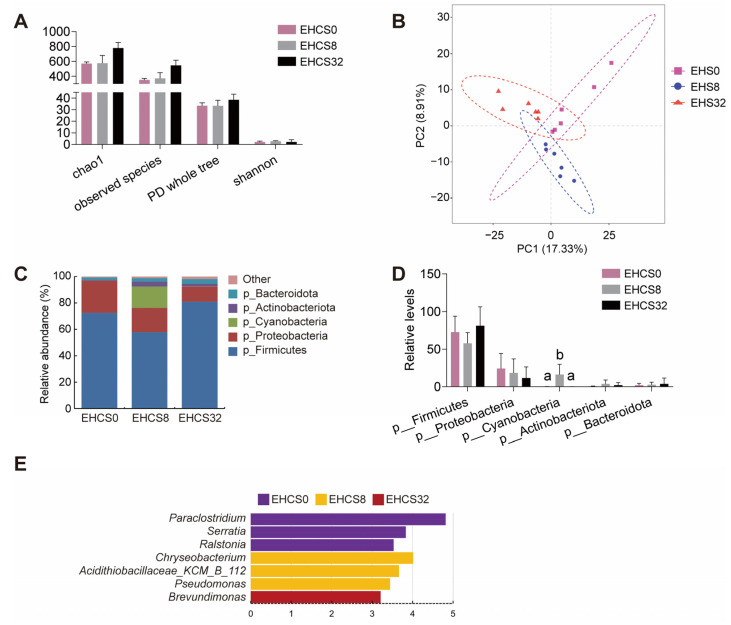
Effects of dietary EHCS inclusion on the intestinal microbiota. (**A**) Alpha diversity of the intestinal flora. Values are shown as mean ± S.D. (**B**) Beta diversity of the intestinal flora. (**C**,**D**) The relative abundance of the intestinal bacterial phyla. Letters (a,b) denote significance at *p* < 0.05. (**E**) The Lefse analysis of the intestinal bacterial genus.

**Table 1 animals-14-03096-t001:** Formulation and proximate composition analysis (% of dry weight basis) of the basal diet.

Ingredients (%)	EHCS0	EHCS8	EHCS16	EHCS24	EHCS32
EHCS	0.00	8.00	16.00	24.00	32.00
White fishmeal	32.00	24.00	16.00	8.00	0.00
Brown fishmeal	40.00	40.00	40.00	40.00	40.00
α-starch	26.00	25.50	25.00	24.50	24.00
Fish oil	0.50	1.00	1.50	2.00	2.50
Choline chloride	0.50	0.50	0.50	0.50	0.50
Ca(H_2_PO_4_)_2_	0.50	0.50	0.50	0.50	0.50
Mineral premix ^a^	0.10	0.10	0.10	0.10	0.10
Vitamin premix ^b^	0.40	0.40	0.40	0.40	0.40
Total	100.00	100.00	100.00	100.00	100.00
Proximate analysis %, dry weight					
Moisture	7.24	7.47	7.68	7.95	7.84
Crude protein	48.93	48.64	48.19	48.21	47.77
Crude lipid	6.72	6.58	6.60	6.45	6.54
Ash	11.80	10.80	9.87	8.63	7.61
Calcium	3.08	2.70	2.30	1.86	1.39
Phosphorus	2.04	1.91	1.71	1.50	1.28
Total energy	18.56	18.65	18.75	18.88	19.07

^a^ Minerals premix (mg kg^−1^diet): Na_2_SeO_3_ 0.89, Cu_2_(OH)_3_Cl 12.05, Ca(IO_3_)_2_ 2.59, FeSO_4_·H_2_O 666.67, MnSO_4_·H_2_O 94.34, CoSO_4_ 3.64, ZnSO_4_·H_2_O 202.89. ^b^ Vitamin premix (IU or mg/kg diet): Vitamin A 3 333.33 IU, VC 33.33 mg, VD_3_ 416.67 IU, VE 15.00 mg, VK_3_ 1.00 mg, VB_1_ 2.67 mg, VB_2_ 5.00 mg, VB_6_ 2.00 mg, VB_12_ 0.01 mg, nicotinic acid 23.33 mg, folate acid 0.33 mg, biotin 0.05 mg, calcium pantothenate 10.00 mg, inositol 33.33 mg.

**Table 2 animals-14-03096-t002:** Amino acid levels of trial diets (% of dry weight basis).

	EHCS0	EHCS8	EHCS16	EHCS24	EHCS32
Essential amino acid (EAA)					
Lysine	3.71	3.63	3.55	3.47	3.38
Methionine	1.29	1.26	1.23	1.20	1.17
Threonine	2.13	2.06	1.99	1.92	1.85
Tryptophan	0.53	0.54	0.54	0.55	0.56
Leucine	3.55	3.50	3.44	3.39	3.34
Arginine	2.72	2.72	2.71	2.71	2.71
Histidine	1.60	1.58	1.56	1.54	1.52
Valine	2.38	2.33	2.28	2.22	2.17
Isoleucine	2.02	2.00	1.98	1.97	1.95
Phenylalanine	1.73	1.75	1.78	1.81	1.84
Total EAA	21.65	21.36	21.07	20.77	20.48
Non-essential amino acid (NEAA)					
Aspartic acid	4.31	4.31	4.31	4.31	4.31
Cysteine	0.47	0.48	0.48	0.49	0.50
Tyrosine	1.72	1.66	1.59	1.52	1.46
Serine	1.93	1.92	1.90	1.89	1.87
Glutamate	6.07	6.18	6.29	6.39	6.50
Proline	1.91	1.90	1.89	1.87	1.86
Glycine	2.96	2.81	2.66	2.52	2.37
Alanine	2.92	2.80	2.69	2.57	2.45
Total NEAA	22.30	22.05	21.81	21.56	21.31

**Table 3 animals-14-03096-t003:** The contents of antinutritional factors in compound soy protein treated with or without enzymatic hydrolysis (mg g^−1^).

	Before Enzymatic Hydrolysis	After Enzymatic Hydrolysis
Trypsin inhibitor factor	1.75	1.30
Glycinin	23.60	<2.00
β-conglycinin	31.90	3.00

**Table 4 animals-14-03096-t004:** Effects of dietary EHCS inclusion on the growth performance of juvenile American eel.

	EHCS0	EHCS8	EHCS16	EHCS24	EHCS32
IBW (g)	26.00 ± 0.01	25.99 ± 0.02	26.01 ± 0.04	26.00 ± 0.01	26.01 ± 0.02
FBW (g)	49.90 ± 0.73 ^c^	52.77 ± 0.83 ^d^	49.96 ± 0.52 ^c^	46.61 ± 0.66 ^b^	42.86 ± 0.81 ^a^
WGR (%)	91.91 ± 2.78 ^c^	103.06 ± 3.29 ^d^	92.06 ± 2.04 ^c^	79.32 ± 2.63 ^b^	64.76 ± 3.04 ^a^
SGR (%/d)	0.93 ± 0.02 ^c^	1.01 ± 0.02 ^d^	0.93 ± 0.02 ^c^	0.84 ± 0.02 ^b^	0.71 ± 0.03 ^a^
FI (g)	32.15 ± 0.78 ^c^	34.32 ± 1.48 ^d^	31.87 ± 0.63 ^c^	30.16 ± 0.26 ^b^	28.08 ± 0.12 ^a^
FE (%)	74.34 ± 0.84 ^c^	78.07 ± 1.29 ^d^	75.14 ± 0.42 ^c^	68.37 ± 2.44 ^b^	60.01 ± 2.87 ^a^
FCR	1.35 ± 0.02 ^a^	1.28 ± 0.02 ^a^	1.33 ± 0.01 ^a^	1.46 ± 0.05 ^b^	1.67 ± 0.08 ^c^
PER (%)	154.42 ± 1.74 ^c^	162.91 ± 2.71 ^d^	157.49 ± 0.87 ^cd^	143.94 ± 5.14 ^b^	126.91 ± 6.08 ^a^
SR (%)	95.00 ± 2.00 ^b^	96.00 ± 3.27 ^b^	96.00 ± 3.27 ^b^	93.00 ± 8.25 ^ab^	87.00 ± 3.83 ^a^

All values are shown as mean ± S.D. (*n* = 4). Letters (a–d) denote significance at *p* < 0.05. IBW, initial body weight; FBW, final body weight; WGR, weight gain rate; SGR, specific growth rate; FI, feed intake; FE, feed efficiency; FCR, feed conversion ratio; PER, protein efficiency ratio; SR, survival rate.

**Table 5 animals-14-03096-t005:** Effects of dietary EHCS inclusion on the body composition of juvenile American eels (wet weight basis).

	EHCS0	EHCS8	EHCS16	EHCS24	EHCS32
Moisture (%)	68.43 ± 1.05	67.99 ± 1.10	69.09 ± 0.61	70.15 ± 2.40	69.24 ± 1.20
Crude protein (%)	17.95 ± 0.64	18.49 ± 0.50	18.06 ± 0.60	17.71 ± 0.25	18.07 ± 0.65
Crude lipid (%)	9.13 ± 0.34	9.04 ± 0.52	9.01 ± 0.30	8.64 ± 0.36	8.68 ± 0.23
Ash (%)	2.06 ± 0.25	1.95 ± 0.13	2.03 ± 0.22	2.12 ± 0.23	2.26 ± 0.07
Calcium (g kg^−1^)	4.35 ± 0.15	4.33 ± 0.17	4.24 ± 0.13	4.18 ± 0.11	4.36 ± 0.19
Phosphorus (g kg^−1^)	3.56 ± 0.50	3.49 ± 0.49	3.64 ± 0.26	3.74 ± 0.43	4.05 ± 0.28

Values are shown as mean ± S.D. (*n* = 4).

## Data Availability

The data that support the findings of this study are available from the corresponding author upon reasonable request.

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
