# Peer review of "Effects of Dietary Inclusion of Enzymatically Hydrolyzed Compound Soy Protein on the Growth Performance and Intestinal Health of Juvenile American Eels (Anguilla rostrata)"

_animals, 2024, doi:10.3390/ani14213096_

Round 1
Reviewer 1 Report
Comments and Suggestions for Authors
This manuscript (animals-3246013) mainly evaluated the application of enzymatically hydrolyzed compound soy protein (EHCS) as a fish meal alternative in the feed of American eels Anguilla rostrata. The experimental design made sense. The statistics was correct. The conclusion was reliable. The presentation of results was a bit long-winded, which should be integrated and more concise. Before publication, the following issues should be addressed.
1. Simple summary or Abstract: The optimal inclusion level of EHSC and the model to quantify this value should be mentioned. Which level should be recommended in the feed of American eel based on the results of this study? The optimal or the maximum level?
2. L67, L71 and other places: provide the latin name of species when they were firstly introduced in the text.
3. L102-103: please justify the ratio of the compound soybean protein and provide the enzymatic hydrolysis details such as temperature, time duration and so on.
4. Experimental design: why the authors did not supplement enough essential animo acids in the EHSC diets up to those of the control?
5. L133-136: details of the sampling procedure should be provided especially about the position of the intestines.
6. Table 4: feeding ratio should be better than feed intake.
7. Figure 6: provide the significant differences for the phyla and genus of the microbiota data. Please justify that the reason only the EHSC8 and EHSC32 fish were analyzed for microbiota composition. L280-282: this sentence should be deleted.
L321: should be 4.1? this subtitle was too long.
8. Discussion: this part was tedious and many sentences were identical of those of the results section. Many results should be integrated and interpretated. For instance, data for the intestinal histology, digestive enzymes and microbiota composition could be discussed together.
Comments on the Quality of English LanguageMinor editing of English language required.
Author Response
Comment 1: This manuscript (animals-3246013) mainly evaluated the application of enzymatically hydrolyzed compound soy protein (EHCS) as a fish meal alternative in the feed of American eels Anguilla rostrata. The experimental design made sense. The statistics was correct. The conclusion was reliable. The presentation of results was a bit long-winded, which should be integrated and more concise. Before publication, the following issues should be addressed.
Response 1: Thank you very much for your good evaluation and constructive comments. We have revised our manuscript based on your important comments. We hope that you are satisfactory to our revisions. Still, if you have any questions, pls. let me know it. Thanks.
Comment 2: Simple summary or Abstract: The optimal inclusion level of EHSC and the model to quantify this value should be mentioned. Which level should be recommended in the feed of American eel based on the results of this study? The optimal or the maximum level?
Response 2: Thanks for your reminder. Based on your important comments, the optimal inclusion level of EHCS, the quantified model, and the recommended level were supplied in Simple Summary of our revised manuscript, as shown below:
Based on the quadratic regression analysis of weight gain rate and feed efficiency against the dietary EHCS inclusion levels, the optimal inclusion levels for American eels ranged from 7.78% to 8.33%, with maximum levels spanning 17.59% to 17.77%, which corresponded to white fishmeal replacement levels of 24.31%–26.03% and 54.97%–55.53%, respectively. To ensure normal growth performance, it is recommended that the inclusion level of EHCS should not exceed 17.77%.
Comment 3: L67, L71 and other places: provide the Latin name of species when they were firstly introduced in the text.
Response 3: Thanks for your reminder. The Latin name of species when they were firstly introduced have been provided in our revised manuscript, as shown below:
Huang et al. [6] demonstrated that replacing up to 45% of dietary fishmeal with enzymatically hydrolyzed soybean did not adversely impact the growth performance of juvenile Chinese mitten crabs (Eriocheir sinensis). Tibaldi et al. [4] also revealed that substituted 50% of dietary fishmeal with enzymatically hydrolyzed soybean meal has no adverse effects on the growth performance and whole-body composition of European sea bass (Dicentrarchus labrax). Moreover, hydrolyzing soybean using a multi-enzyme strategy exhibits a higher efficacy in substituting fishmeal in aquafeeds compared to its single-enzyme counterpart [5]. It was reported that protease-treated soybean meal could substitute for 20% of fishmeal in the diets of largemouth bass (Micropterus salmoides), whereas a combination treatment with protease and non-starch polysaccharide enzymes could facilitate soybean meal replaced up to 47.27% of dietary fishmeal for largemouth bass [5].
Comment 4: L102-103: please justify the ratio of the compound soybean protein and provide the enzymatic hydrolysis details such as temperature, time duration and so on.
Response 4: Thanks for your reminder. We have justified the ratio of the compound soybean protein and supplemented the details of enzymatic hydrolysis in our revised manuscript, as shown below:
The EHCS comprised a mixture of soybean meal, soy protein concentrate, and soy protein isolate in a 4:5:1 ratio, which was enzymatically hydrolyzed with alkaline protease (750 IU/g substrate), neutral protease (250 IU/g substrate), and acid protease (250 IU/g substrate) at a controlled temperature of 40℃ for a duration of 24 hours.
Comment 5: Experimental design: why the authors did not supplement enough essential amino acids in the EHSC diets up to those of the control?
Response 5: Thanks for your question. The optimal amino acid requirement for American eels remains unclear, and it was previously uncertain whether the amino acid composition of the control diet met the requirement of American eels. Therefore, we did not supplement additional amino acids in the EHCS diets for this study. As a matter of fact, our findings suggested that the amino acid composition of the EHCS8 diet might be more suitable for American eels compared to that of the control diet. Based on this study, we have initiated a new study that explore whether supplementing additional amino acids in the EHCS can enhance its applicability in the diet of American eels, with the forthcoming results expected to yield valuable insights.
Comment 6: L133-136: details of the sampling procedure should be provided especially about the position of the intestines.
Response 6: Thanks for your reminder. The details of the sampling procedure have been improved in Section 2.3 of the Materials and Methods section in our revised manuscript based on your suggestion, as shown below:
At the end of the feeding trial, the eels were fasted for 24 hours. The number and weight of eels in each tank was recorded. Eels were anesthetized using eugenol in a mixture of 50 mg L-1at a ratio of 1:4 for about 5 min. The collected serum samples were stored at -80°C for subsequent biochemical analysis. Anterior intestine samples from four eels per tank were sampled and rapidly frozen in liquid nitrogen and stored at -80°C for determining intestinal digestive enzyme activity and antioxidant-related indices. Anterior intestine samples from two eels per tank were fixed in Bouin's solution and stored at 4°C for morphological observation. The anterior intestines of six eels per tank were rapidly frozen in liquid nitrogen and then stored at -80°C for intestinal microbiota analysis. Ten eels from each tank were randomly selected for whole-body composition analysis.
Comment 7: Table 4: feeding ratio should be better than feed intake.
Response 7: Thanks for your suggestion. We have supplemented feeding ratio in Table 4 and Section 3.1 of the Results section in our revised manuscript. For details, pls see our revised manuscript.
Comment 8: Figure 6: provide the significant differences for the phyla and genus of the microbiota data.
Response 8: The significant differences for the phyla of the microbiota data have been provided in Figure 6 of our revised manuscript. For details, pls see our revised manuscript. Lefse analysis is restricted to genera exhibiting statistically significant differences. Therefore, the genera identified through Lefse analysis in this study are those characterized by statistically significant differences.
We hope the revisions would make you satisfied. Still, if you have any questions, pls. do not hesitate to contact me. Thanks.
Comment 9: Please justify that the reason only the EHSC8 and EHSC32 fish were analyzed for microbiota composition.
Response 9: Thanks for your reminder. We have supplemented the reason that only the EHSC0, EHSC8, and EHCS32 fish were analyzed for microbiota composition in Section 3.7 of the Results section in our revised manuscript, as shown below:
Results
Considering that American eels fed the EHCS8 diet exhibited superior growth performance and intestinal health, while those fed the EHCS32 diet experienced growth arrest and intestinal injury, compared to those fed the EHCS0 diet, an analysis of intestinal microbiota was carried out among American eels fed these three diets to elevated whether different dietary EHCS inclusion levels affect the microbiota composition of American eels.
Comment 10: L280-282: this sentence should be deleted.
Response 10: Thanks for your reminder. We have deleted this sentence in our revised manuscript.
Comment 11: L321: should be 4.1? this subtitle was too long.
Response 11: Thanks for your reminder. We have reordered the subheadings and shorten the subtitle of 4.1, as shown below:
4.1. Growth performance
Comment 12: Discussion: this part was tedious and many sentences were identical of those of the results section. Many results should be integrated and interpretated. For instance, data for the intestinal histology, digestive enzymes and microbiota composition could be discussed together.
Response 12: Thanks for your suggestions. We have improved our Discussion to enhance its clarity and conciseness based on your comments. For details, pls see our revised manuscript.
Reviewer 2 Report
Comments and Suggestions for Authors
Thank you for this broad study. However, there are some things that may need more attention.
In line, 131 please mention how much of anesthetics was used and for how long.
And for the rearing of the animals: was it a flow through system or a static system? How often were the water parameters monitored? With an automatic system or manually?
And the lines 280 to 282 can be deleted.
And for the beta diversity analysis: the PC1 and PC2 have only low percentages. Is this typical? How many dimensions did the analyses have?
Comments on the Quality of English LanguageSome sentences should be checked, but most of the manuscript is okay.
Author Response
Comment 1: Thank you for this broad study. However, there are some things that may need more attention.
Response 1: Thank you very much for your favorable appraisal of our manuscript. We have strictly revised our manuscript based on your important comments. Also, we provide our responses, on a point-to-point basis, on your important comments. For details, please see the text.
Comment 2: In line, 131 please mention how much of anesthetics was used and for how long.
Response 2: Thanks for your reminder. The details of anesthetics used in this study have been improved in Section 2.3 of the Materials and Methods section in our revised manuscript based on your suggestion, as shown below:
At the end of the feeding trial, the eels were fasted for 24 hours. The number and weight of eels in each tank was recorded. Eels were anesthetized using eugenol in a mixture of 50 mg L-1at a ratio of 1:4 for about 5 min.
Comment 3: And for the rearing of the animals: was it a flow through system or a static system? How often were the water parameters monitored? With an automatic system or manually?
Response 3: A flow-through system was employed in this study and the water parameters were monitored manually twice a week. This information has been supplemented in our revised manuscript, as shown below:
An automatic temperature control system with flow-through was utilized throughout the trial.
The water quality parameters were monitored manually twice a week and are detailed as follows.
Comment 4: And the lines 280 to 282 can be deleted.
Response 4: Thanks for your reminder. We have deleted this sentence in our revised manuscript.
Comment 5: And for the beta diversity analysis: the PC1 and PC2 have only low percentages. Is this typical? How many dimensions did the analyses have?
Response 5: Yes, and this low percentages of PC1 to PC2 are frequently observed in the beta diversity analysis across numerous experimental trials [1-6]. The beta diversity in this study was analyzed based on partial least squares discrimination analysis (PLS-DA) which is a supervised statistical method of discriminant analysis that different from PCA. PLS-DA analysis is based on two datasets, X and Y, which correspond to two multidimensional spaces.
References
[1] Sun, S.; Gong, C.; Deng C.; Yu, H.; Zheng, D.; Wang, L.; Sun, J.; Song, F.; Luo, J. Effects of salinity stress on the growth performance, health status, and intestinal microbiota of juvenile Micropterus salmoides. Aquaculture 2023. 576, 739888.
[2] Chen, B.; Qiu, J.; Wang, Y.; Huang, W.; Zhao, H.; Zhu, X.; Peng, K. Condensed tannins increased intestinal permeability of Chinese seabass (Lateolabrax maculatus) based on microbiome-metabolomics analysis. Aquaculture 2022. 560, 738615.
[3] Rui, A.; Sanahuja, I.; Andree, K.B.; Furones, D.; Holhorea, P.G.; Calduch-Giner, J.A.; Pastor, J.J.; Vinas, M.; Perez-Sanchea, J.; Morais, S.; Gisbert, E. Genotype x nutrition interactions in European sea bass (Dicentrarchus labrax): Effects on gut health and intestinal microbiota. Aquaculture 2024. 581, 740378.
[4] Lokesh, J.; Ghislain, M.; Reyrolle, M.; Bechec, M.L.; Pigot, T.; Terrier, F.; Roy, J.; Panserat, S.; Ricaud, K. Prebiotics modify host metabolism in rainbow trout (Oncorhynchus mykiss) fed with a total plant-based diet: Potential implications for microbiome-mediated diet optimization. Aquaculture 2022. 561, 738699.
[5] Obianwuna, U.E.; Huang, L.; Zhang, H.; Wang, J.; Qi, G.; Qiu, K.; Wu, S. Fermented soybean meal improved laying performance and egg quality of laying hens by modulating cecal microbiota, nutrient digestibility, intestinal health, antioxidant and immunological functions. Anim Nutr 2024. 18, 309-321.
[6] Qin, W.; Xu, B.; Chen, Y.; Yang, W.; Xu, Y.; Huang, J.; Duo, T.; Mao, Y.; Zhou, G.; Yan, X.; Ma, L. Dietary ellagic acid supplementation attenuates intestinal damage and oxidative stress by regulating gut microbiota in weanling piglets. Anim Nutr 2022. 11, 322-333.
Comment 6: Some sentences should be checked, but most of the manuscript is okay.
Response 6: Thanks for your reminder. We have thoroughly reviewed our manuscript and rectified errors in accordance with your suggestions.
Reviewer 3 Report
Comments and Suggestions for Authors
This study investigated the effects of substituting dietary white fishmeal with enzymatically hydrolyzed compound soy protein (EHCS), treated with a combination of multiple enzymes, on the growth performance and intestinal health of juvenile American eels (Anguilla rostrata). The results have reference value for guiding practical feed production.
However, there is obvious flaw in the experimental design. In addition to the use of white fishmeal, the experimental feed formula also included brown fishmeal. Although the content of brown fish meal is unchanged, its high content will certainly have a significant impact on the proportion and effect of replacement fishmeal. The results do not support the conclusion. Secondly, the conditions of enzymatic hydrolysis of soybean protein, such as enzyme concentration, time and temperature, were not fully explained. and the conclusions could not support the results.
For these main reasons, the manuscript could not be recommended for publication unless the authors rewrote the manuscript.
Comments on the Quality of English LanguageThe context is necessary to be further revised in quality of English language.
Author Response
Comment 1: This study investigated the effects of substituting dietary white fishmeal with enzymatically hydrolyzed compound soy protein (EHCS), treated with a combination of multiple enzymes, on the growth performance and intestinal health of juvenile American eels (Anguilla rostrata). The results have reference value for guiding practical feed production.
Response 1: Thank you very much for your favorable appraisal of our work.
Comment 2: However, there is obvious flaw in the experimental design. In addition to the use of white fishmeal, the experimental feed formula also included brown fishmeal. Although the content of brown fish meal is unchanged, its high content will certainly have a significant impact on the proportion and effect of replacement fishmeal. The results do not support the conclusion. Secondly, the conditions of enzymatic hydrolysis of soybean protein, such as enzyme concentration, time and temperature, were not fully explained. and the conclusions could not support the results. For these main reasons, the manuscript could not be recommended for publication unless the authors rewrote the manuscript.
Response 2: Here, let me answer your concerns one by one:
(1) American eels have a high protein requirement, with premium-grade white fishmeal traditionally serving as the primary protein source in their diets. White fishmeal is the principal dietary expense for American eels. To our knowledge, this is the first investigation into fishmeal substitution in American eel diets. Although a certain proportion of brown fishmeal remains in the diet, our study provides a critical basis for supporting the reduction of white fishmeal use in American eel diets, thereby lowering feed costs. Our findings also offer insights into further reducing white fishmeal usage in American eel diets, addressing factors such as amino acid imbalance, antinutritional factors, and the role of intestinal damage as a primary limitation in fishmeal substitution in American eels. Therefore, this study is of great significance for reducing the reliance of American eel diet on white fishmeal and simultaneously promoting the growth performance and overall health of American eels.
To avoid our conclusions being misunderstood by readers, we have emphasized that this study mainly explored white fishmeal substitution in our revised manuscript. For details, pls see our revised manuscript.
(2) The conditions of enzymatic hydrolysis of soybean protein, such as enzyme concentration, time and temperature, have been supplemented in our revised manuscript. For details, pls see our revised manuscript.
We hope the revisions would make you satisfied. Still, if you have any questions, pls. do not hesitate to contact me. Thanks.
Comment 3: The context is necessary to be further revised in quality of English language.
Response 3: Thanks for your suggestion. We have thoroughly reviewed our manuscript and rectified errors in accordance with your suggestions.
Round 2
Reviewer 1 Report
Comments and Suggestions for Authors
The authors have well addressed my concerns. Thus, this version of the manuscript is acceptible.
Reviewer 3 Report
Comments and Suggestions for Authors
The authors revised the manuscript well and it can be considered for publication.